# Adaptive Gating in Mixture-of-Experts based Language Models

**Jiamin Li[1], Qiang Su[1], Yitao Yang[2], Yimin Jiang, Cong Wang[1], Hong Xu[2]**
[1]City University of Hong Kong, [2]The Chinese University of Hong Kong
jiamin.li@my.cityu.edu.hk, qiang.su@my.cityu.edu.hk
ytyang@cse.cuhk.edu.hk, jymthu@gmail.com
congwang@cityu.edu.hk, hongxu@cuhk.edu.hk

## Abstract

Large language models, such as OpenAI's Chat-GPT, have demonstrated exceptional language understanding capabilities in various NLP tasks. Sparsely activated mixture-of-experts (MoE) has emerged as a promising solution for scaling models while maintaining a constant number of computational operations. Existing MoE model adopts a fixed gating network where each token is computed by the same number of experts. However, this approach contradicts our intuition that the tokens in each sequence vary in terms of their linguistic complexity and, consequently, require different computational costs. Little is discussed in prior research on the trade-off between computation per token and model performance. This paper introduces adaptive gating in MoE, a flexible training strategy that allows tokens to be processed by a variable number of experts based on expert probability distribution. The proposed framework preserves sparsity while improving training efficiency. Additionally, curriculum learning is leveraged to further reduce training time. Extensive experiments on diverse NLP tasks show that adaptive gating reduces at most 22.5% training time while maintaining inference quality. Moreover, we conduct a comprehensive analysis of the routing decisions and present our insights when adaptive gating is used.

## 1 Introduction

The field of natural language processing (NLP) has undergone a remarkable revolution driven by the rapid advancements in language models (Cha; Touvron et al., 2023; Bar; pal). They exhibit so-called "emergent" capabilities for a wide variety of applications (Wei et al., 2022). However, as demands for these applications continue to grow, scalability of these models poses an increasingly challenging hurdle due to constraints in computational resources, memory capacity, interconnect bandwidth, etc. (Pope et al., 2023).

Sparsely-activated mixture-of-experts (MoE) is a promising paradigm to address the scalability issue while maintaining a constant number of computation FLOPs (Lepikhin et al., 2020; Fedus et al., 2021). MoE utilizes an ensemble of experts to collectively tackle the learning task. Each input activates a subset of experts, resulting in a dynamically-changing and sparse computation graph. This method effectively distributes the computation among experts, increases model capacity and improves training efficiency (Du et al., 2022; Rajbhandari et al., 2022). Very recently, there has been quite some prior work on improving the performance of Transformers using MoE (Rajbhandari et al., 2022; Zoph et al., 2022; Chen et al., 2023a; Gale et al., 2022).

Despite MoE's benefit in scalability, it suffers from suboptimal training efficiency. In particular, we focus on the gating mechanism that selects the experts for each token in this work. Existing MoE models adopt a fixed top-2 gating in training while employing top-1 gating during inference for shorter response times. Top-2 gating entails twice the computational cost per token and doubles the data transfer size of all-to-all operations compared to top-1. Yet, it remains unclear whether top-2 gating actually leads to performance gains that could justify the additional overheads. Therefore, a comprehensive analysis of the trade-off between training efficiency and model performance is increasingly crucial. More practically, how to construct an MoE language model that effectively balances training efficiency and performance, is of great interest and imminent value.

Towards this end, we present our first attempt to empirically characterize and improve the efficiency of the gating mechanism in MoE. We observe that across various models and tasks, a large number of tokens display simple linguistic characteristics or a single dominant feature, which allows them to be effectively processed using just the top-1 expert.

This observation suggests that the current top-2 gating strategy incurs unnecessary computation costs for a significant number of tokens.

Motivated by this insight, we further introduce adaptive gating in MoE that enables tokens to be processed by a *flexible* number of experts depending on the gating decision. Our approach, in contrast to conventional MoE models, preserves the sparsity of MoE models while enhancing flexibility in token handling. We incorporate a threshold within the gating network to conduct adaptive token routing based on the distribution of expert probabilities. With adaptive gating, the majority of tokens use simple top-1 gating; top-2 gating is selectively applied only when necessary and beneficial, thus significantly reducing the computation cost. However, the training efficiency cannot achieve the same improvement as the computation cost due to the fact that tokens with top-2 gating always incur a longer training step, thus becoming the bottleneck. Therefore, to enhance training efficiency even further, we leverage the idea of curriculum learning by strategically adjusting the order of training data samples.

We conduct extensive experiments on six NLP tasks with different encoder and decoder models. The results show that our approach can effectively reduce the end-to-end training time by at most 22.5%, while achieving comparable inference quality with top-2 gating MoE models. Moreover, we show that the tokens routed to two experts are coupled with the nature of each NLP task. For sentiment analysis, those are the tokens expressing neutral opinions; translation task pays attention to sentences with complex structure; Question and Answer connects key words in question and context and assign both with top-2 gating; summarization puts more effort in understanding the pronouns and finding tokens expressing central idea; top-2 routing decision changes along with the token to generated in text completion task and conversational tokens in dialogue response task use top-2 experts frequently. Empirically, we find that a small threshold value (i.e. 0.1, 0.2) in adaptive gating can lead to a similar performance as top-2 gating.

Our contributions are as follows:

- We propose adaptive gating in the MoE training scheme, which enables tokens to be processed by a flexible number of experts.
- We leverage curriculum learning to alleviate the training bottleneck caused by varying execution times of tokens.

- We conduct extensive experiments on various NLP tasks and datasets and present a thorough analysis of the gating decision of the tokens to prove the effectiveness and efficiency of adaptive gating.

## 2 Background

### 2.1 Mixture-of-Experts

Mixture-of-Experts (MoE) has been adopted in various deep neural network models (Shen et al., 2023; Chen et al., 2023b) and has shown great promise in enhancing the performance of language models. For example, GShard (Lepikhin et al., 2020) and Switch Transformer (Fedus et al., 2021) effectively scale Transformer-based language models with MoE layers.

In particular, these models typically employ an MoE layer to substitute the feed-forward network (FFN) layer. The MoE layer comprises multiple FFNs, each acting as an expert, along with a gating network. Each expert $i$ is a fully-connected two-layer network utilizing ReLU activation and with its own set of parameters. For a given token $x$, the output of an expert can be defined as:

$$FFN_i(x) = ReLU(x \cdot W_0^i) \cdot W_1^i, \qquad (1)$$

where $W_0^i$ and $W_1^i$ are the trainable weights of the two linear layers in expert $i$.

The gating network takes in the embedding vector of each token $x$ and multiplies them with its trainable matrix $W_G$. The gate value for a specific token can be determined through:

$$R = softmax(x \cdot W_G). \qquad (2)$$

This softmax activation $R$ indicates the weight of each expert in processing the token. The gating network then dispatches this token to top-$k$ experts with $k$ highest activations. The final output of the MoE layer is:

$$y = \sum_{i \in E} R_i \cdot FFN_i(x), \qquad (3)$$

that is, the weighted sum of outputs from selected expert(s) $E \subset \{FFN_1, FFN_2...FFN_N\}$. The sparse nature of MoE improves the model scaling in size without increasing the training cost.

**Related work.** Several prior works have explored the efficient use of gating or expert selection in MoE. Aoki et al., 2022; Zhou et al., 2022; Hazimeh et al., 2021; Ma et al., 2018 propose different

approaches to encourage expert specialization. Dai et al., 2022 adopt a pre-defined expert assignment for each input categories. Roller et al., 2021; Zuo et al., 2021 propose to remove gating networks. Zhou et al., 2022 present a novel selection mechanism where experts selects token instead of token selecting experts. Hazimeh et al., 2021 introduce multiple routing policies to enhance specialization in multi-task scenario. Roller et al., 2021 use deterministic hashing, while Zuo et al., 2021 use stochastic routing. However, it could lead to inconsistent inference results. Therefore, they employ a regularized loss to penalize the discrepancy of expert selection. All existing work adopts a fixed and equal computation capacity for each token and expert, while we look into the trade-off between computation costs and model performance with adaptive gating.

## 3 Design

We now discuss the design of adaptive gating in MoE for training.

### 3.1 Adaptive Gating in MoE

**Observation.** We first present our empirical findings from experiments with classical MoE models. Specifically, we extract the softmax activations and analyze the probability distribution of expert selection for each token in the gating network. Figures 1 depict the normalized activation values of four sampled tokens across 16 experts. We see that for tokens 1 and 4, their activations of the top-1 and top-2 expert are very close as shown in Figures 1a and 1d, while for tokens 2 and 3 a significant bias towards the top-1 expert exists as in Figures 1b and 1c. We find that these significantly-biased distribution accounts for at least 55% of all the tokens in our evaluation.

**Adaptive gating.** Previous work has demonstrated that MoE experts specialize in different linguistic aspects. Building upon our empirical findings, one can see that many tokens can be effectively handled by a single expert during the training stage. To control the number of experts handling each token, we introduce a threshold parameter, denoted as $T$. If the activation value difference between the top-1 expert, denoted as $i$, and the top-2 expert, denoted as $j$, is within the threshold $T$, we consider the token as requiring both expert $i$ and expert $j$ for processing. Otherwise, we route the token only to the top-1 expert.

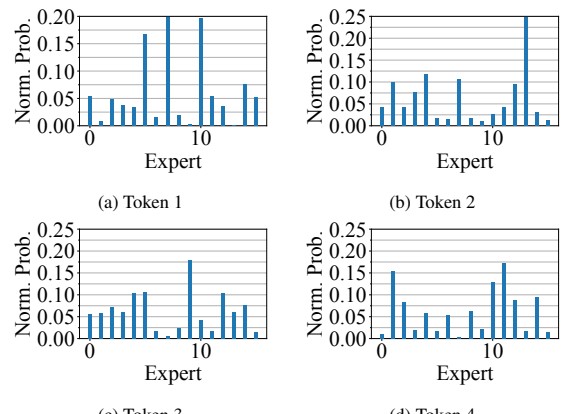

(a) Token 1      (b) Token 2

(c) Token 3      (d) Token 4

Figure 1: Normalized expert probability computed by top-2 gating network from four sampled tokens. Here we use the Sentiment analysis task list in Table 2.

| Gate | Norm. Computation | Norm. MoE Layer Running Time |
|------|-------------------|------------------------------|
| Top-1 | 0.5 | 0.67 |
| Adaptive (80% Top-1) | 0.6x | 0.76x |
| Adaptive (50% Top-1) | 0.75x | 0.92x |
| Adaptive (20% Top-1) | 0.9x | 0.97x |

Table 1: We compare the computation savings and running time reduction of the MoE layer of varying degrees of top-1 gating against top-2 gating. The MoE layer running time is measured on our testbed Section 4.3. Tokens are randomly selected from the data batch. Here we also use the Sentiment analysis task list in Table 2. We show the results averaged from 40 runs.

**Load balancing loss.** Adaptive gating uses a flexible number of experts to process each token. This flexibility, however, adds extra difficulty to the load balancing problem in training which aims to evenly distribute tokens among all experts. As it is still important to prevent the gating network from overly concentrating on a very small number of experts, in adaptive gating, we impose the soft load balancing constraints on the top-1 gating decisions, while allowing top-2 gating decisions to be trained without any soft constraints. That is, the loss of each MoE layer $i$ becomes:

$$L_i = E_i \sum_{e \in E} f_e^1 p_e, \qquad (4)$$

where $f_e^1$ is the fraction of tokens dispatched to expert $e$ among those processed by top-1 gating; $p_e$ is the average gating probability to expert $e$ over all tokens in the current batch, and $E_i$ is the number of experts at layer $i$ just as in classical MoE (Fedus et al., 2021).

### 3.2 Batching

**Challenge.** While adaptive gating provides effective computational savings, Transformer MoE's model architecture poses a significant challenge to training efficiency. Specifically, there is a mismatch

| Task | Dataset | Model | Architecture |
|------|---------|-------|--------------|
| Sentiment analysis | SST-2 (Socher et al., 2013) | BERT-Base (Devlin et al., 2018) | 12-layer encoder |
| Translation | WMT19 (De->En) (Foundation) | FSMT (Ng et al., 2020) | 6-layer encoder, 6-layer decoder |
| Question and Answer | SQuAD (Rajpurkar et al., 2016) | BERT-Base (Devlin et al., 2018) | 12-layer encoder |
| Summarization | CNN/Daily Mail (Hermann et al., 2015; See et al., 2017) | BART-Large (Lewis et al., 2019) | 12-layer encoder, 12-layer decoder |
| Text generation | wikitext (Merity et al., 2016) | GPT-2 (Radford et al., 2019) | 24-layer decoder |
| Dialogue response | SODA (Kim et al., 2022) | DialoGPT-medium (Zhang et al., 2020) | 24-layer decoder |

Table 2: Overall performance of adaptive MoE and compared baselines in different NLP tasks. All the models converge to the same loss value.

in the data processing granularity between the MoE experts and the Attention layer. The MoE experts operate on individual tokens, while the Attention layer requires input in the form of a complete sentence. As a result, although the processing time for a large portion of tokens is reduced by half in the MoE layer, we still need to wait until the remaining tokens (in the same data batch) complete their top-2 processing. Consequently, training step time cannot enjoy the same reduction as in computation. Table 1 shows the computation reduction as well as empirical MoE layer running time, both normalized to conventional top-2 gating. We use PyTorch Profiler to obtain the computation time of MoE layer. For simplicity, here we force a fixed percentage of tokens to be routed to only top-1 expert and measure the running time. The reduction in running time is clearly much smaller than the computation savings.

**Curriculum learning.** In adaptive gating, we propose to incorporate the concept of curriculum learning to address the aforementioned training efficiency challenge. Curriculum learning (Bengio et al., 2009), as the name implies, is a paradigm where training examples are presented to a model in increasing order of complexity. It aims to enhance the learning efficiency and generalization performance of models. By carefully designing the curriculum, the model is exposed to easier examples at the initial stages, allowing it to build a solid foundation before tackling more challenging concepts. This gradual learning process has shown promising results in NLP (Wang et al., 2021).

**Adjust training data order.** Our intuition is that the number of experts required by each token can be an indicator of the token complexity. We can therefore reorder the training data in a way that prioritizes simpler sequences during model training. Additionally, we can group together training data with similar complexity levels to minimize the bottleneck effect caused by difficult tokens in need of top-2 experts.

To quantify the complexity of a training sample

$d$, we define a complexity vector $C$:

$$C_d = [r_0^d, r_1^d, ... r_L^d], \quad (5)$$

where $L$ is the number of MoE layers in the model, and $r_i$ represents the ratio of tokens processed by top-2 experts to the sequence length (i.e., the total number of tokens in data sample $d$) in layer $i$.

To determine the order of the training data, we identify the data sample with the fewest tokens processed by top-2 experts, and calculate the cosine similarity using complexity vectors of the remaining data samples. Training data is then reordered based on this similarity value, starting from the most similar ones. This approach allows the model to gradually learn from simpler sequences and progressively handle more complex sequences.

## 4 Evaluation

We evaluate adaptive gating in MoE on six NLP tasks using various encoder and decoder models. We then analyze the gating decision to better understand the effectiveness of adaptive gating.

### 4.1 Tasks and Models

Table 2 summarizes the details.

### 4.2 Baselines

We use the Transformer models from HuggingFace and convert the FFN layers to MoE layers (Komatsuzaki et al., 2022). We compare adaptive gating's training efficiency with the following three baselines and then evaluate the inference performance with top-1 gating MoE.

**Dense models.** Transformer with no MoE layers.

**Top-2 gating MoE.** MoE models with top-2 gating (Lepikhin et al., 2020; Hazimeh et al., 2021) for training.

**Top-1 gating MoE (Switch Transformer).** Switch Transformer (Fedus et al., 2021; Kim et al., 2021; Xue et al., 2022) uses top-1 gating to mitigate training instabilities.

| Task | Scheme | Norm. Training Time | Computation FLOPs | Inference Performance |
|------|--------|---------------------|-------------------|------------------------|
| Sentiment analysis | Dense | 0.88x | 2.18G | 0.912 |
| | Top-2 Gating | 1x | 3.28G | 0.918 |
| | Top-1 Gating | 0.99x | 2.18G | 0.902 |
| (Accuracy) | Adaptive Gating | 0.77x | 2.30G | **0.919** |
| En->De translation | Dense | 0.87x | 10.6G | 40.9 |
| | Top-2 Gating | 1x | 15.9G | **41.1** |
| | Top-1 Gating | 1.04x | 10.6G | 39.5 |
| (BLEU Score) | Adaptive Gating | 0.79x | 11.5G | **41.1** |
| Question and Answer | Dense | 0.84x | 2.18G | 75.7 |
| | Top-2 Gating | 1x | 3.27G | **77.6** |
| | Top-1 Gating | 1.07x | 2.18G | 75.5 |
| (F1 Score) | Adaptive Gating | 0.86x | 2.36G | 77.4 |
| Summarization | Dense | 0.89x | 79G | 42.3 |
| | Top-2 Gating | 1x | 119G | **43.4** |
| | Top-1 Gating | 1.06x | 79G | 40.8 |
| (ROUGE-1) | Adaptive Gating | 0.86x | 87G | 43.3 |
| Text completion | Dense | 0.84x | 3.4T | 16.3 |
| | Top-2 Gating | 1x | 4.9T | **17.8** |
| | Top-1 Gating | 1.14x | 3.4T | 16.5 |
| (Perplexity) | Adaptive Gating | 0.89x | 3.73T | 17.5 |
| Dialogue response | Dense | 0.82x | 3.4T | 12.5 |
| | Top-2 Gating | 1x | 4.9T | **13.4** |
| | Top-1 Gating | 0.93x | 3.4T | 12.6 |
| (Perplexity) | Adaptive Gating | 0.82x | 3.76T | 13.3 |

Table 3: Overall performance of adaptive gating and compared baselines in different NLP tasks. We normalize the training time with reference to the performance of top-2 gating MoE. All the schemes in the same task converge to the same loss.

## 4.3 Training Configurations

We use 8 A100 GPUs, each with 40 GB memory. Data and expert parallel is used for distributed training. We distribute the experts evenly among all the GPUs. In terms of hyperparameters and model architecture, we adopt the default configurations established in the existing models (Wolf et al., 2020; Kwon and Chung, 2023).

**Model architecture.** BERT-Base has 12 attention heads per layer. The hidden size is 768 and the intermediate dimension is 3072. The Transformer model has 16 attention heads. The hidden size is 1024 and the intermediate dimension in encoder and decoder layers are 8192 and 4096, respectively. BART-Large has 16 attention heads. The hidden size is 1024 and the intermediate dimension is 4096. GPT-2 and DialoGPT-medium have 16 attention heads. The hidden size is 1024 and the intermediate dimension is 4096.

**Hyperparameters.** BERT-Base has a batch size of 24 and the learning rate is 0.00003. The maximum number of tokens for the translation model is 4096 with a learning rate of 0.0005. The maximum number of tokens allowed for BART-Large is set to 4096. The learning rate is 0.00001. The batch size of GPT-2 is 8 with a learning rate of 0.00015.

For DialoGPT-medium, the batch size and learning rate are 64 and 0.0001.

**MoE configurations.** The parameter size of the FFN in each model is the same in Baseline and MoE models and we set the number of FFNs (i.e. experts) to 16 for all evaluated tasks. The coefficient of the load balancing loss is 0.01. No capacity constraints are enabled so no tokens would be dropped. The expert parameters are randomly initialized. We normalize the expert probability in adaptive gating and set the threshold $T$ to 0.1.

## 4.4 Overall Performance

We present the overall training and inference performance in Table 3. Overall, adaptive gating achieves comparable performance to the baselines while significantly reducing the training time even compared to top-1 gating. This is because though top-1 gating maximizes the computation saving, it makes training more difficult to converge to the same loss value, eventually leading to slightly longer training time compared to top-2 gating in 4 out of 6 tasks we run. An in-depth analysis of how adaptive gating works in connection to each task is presented in Section 4.5.

**Sentiment analysis.** Adaptive gating in MoE outperforms both Dense models and top-2 gating MoE

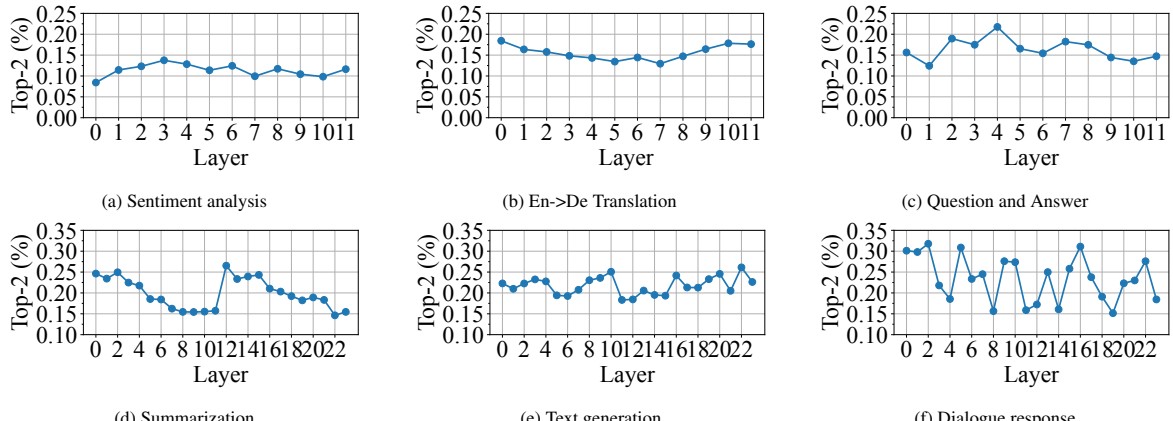

Figure 2: Percentage of tokens computed by top-2 experts over all the tokens in each layer when using adaptive gating in MoE.

in all metrics. While the average computation FLOPs per token is higher with adaptive gating compared to top-1 gating MoE, which represents the minimum possible FLOPs in the MoE structure, adaptive gating requires less training time and achieves superior accuracy during the inference stage. This is consistent across all the tasks. Notably, only 11.3% of the tokens in our evaluation receive two experts, which is the lowest among all tasks. Compared to top-2 gating, adaptive gating focuses on assigning more experts to tokens that represent neutral opinions, allowing for a more comprehensive decision-making process. Conversely, tokens expressing little or obvious sentiment are given less attention without degrading accuracy.

**Translation.** Adaptive gating delivers the same performance with top-2 gating while reducing training time and FLOPs per token by 25.6% and 38.2%, respectively. Notably, we observe that the gating network in adaptive gating exhibits a particular focus on the complexity of sentence structures. Even tokens that appear linguistically simple can involve two experts when they appear in sentences with intricate structures and grammar. Overall, 25.6% of all trained tokens are routed to two experts.

**Question and Answer.** The training time with adaptive gating is 85.7% that of top-2 gating. Although its inference performance is slightly lower, it still outperforms top-1 gating. Through our experiments (refer to Section 4.6), we discover that adaptive gating achieves the best results when the threshold is set to 0.2 for Question and Answer. The gating decision is influenced by both the context and the specific question being asked. For this task 16.4% tokens receive top-2 processing.

**Summarization.** Summarization is the most challenging task in our evaluation, as it involves processing long and information-rich articles. Adap-

tive gating takes 11.8% less training time than top-2 gating. However, its inference performance slightly lags behind. Particularly, in adaptive gating tokens selected for top-2 experts exhibit significant variations across different layers. We provide a more detailed analysis of this observation in Section 4.5.

**Text completion.** We use a GPT-like decoder-only architecture for this task. Adaptive gating achieves similar performance as top-2 gating and Dense models while outperforming top-1 gating. When compared to top-2 gating, only 21.8% tokens rely on two experts, resulting in a reduction of 23.8% in average computation FLOPs per token. The selection of tokens utilizing two experts varies considerably due to the diverse nature of the input.

**Dialogue response.** Dialogue response requires more nuanced processing compared to simple text generation, as it involves generating responses in a targeted role based on narrative input and dialogue history. The sparsity introduced by MoE is advantageous for this task. All three MoE approaches outperform the Dense model. Among all the tasks evaluated, dialogue response exhibits the highest percentage, 23.4% of tokens routed to two experts, indicating the higher utilization of the top-2 gating mechanism among all the tasks. Upon evaluating the tokens, we observe that this task can be viewed as a combination of all the other evaluated tasks.

## 4.5 Analysis and Insights

While it is intuitive to understand that some minor tokens (e.g., "a", "the", "is") only need top-1 expert to process, this does not fully explain how and why adaptive gating works in different NLP tasks. Thus we analyze how the tokens are processed in training with adaptive gating, and make quite a few interesting observations that can help better answer this question. In a broader sense, we believe

our insights are also instrumental towards building better language models.

Note that when BPE tokenizer is used, we aggregate the result by mapping the tokens to the natural language word and perform analysis on the aggregated statistics.

**Sentiment analysis.** Sentiment analysis exhibits the lowest percentage of top-2 gating among all tasks, and the percentage is stable across layers (Figure 2a). The top-2 gating mechanism focuses on two main types of input here. First, it frequently selects tokens that express a more neutral opinion since they are more difficult to classify (Table 4). Second, tokens associated with sarcastic statements, double negatives, or conflicting opinions are also commonly routed to two experts. Adaptive gating effectively identifies these tokens early on in the model as they are relatively easy to extract, which explains the stable percentage across layers. A special case is when the input does not explicitly convey any sentiment. Adaptive gating tends to initially route all tokens to either the top-1 or top-2 experts and gradually narrows down to more informative tokens. A typical instance of this is "as a dentist's waiting room."

**Translation.** We focus on English-to-German translation only. We examine the top-2 gating results based on our understanding of the source sentences. The distribution of the top-2 gating percentages varies between the encoder and decoder layers, exhibiting a gradual decrease in the encoder layers and an increase in the decoder layers (Figure 2b). From sampled tokens and the adjusted data order in adaptive gating, we observe that tokens requiring two experts are usually within the same sentence. This observation leads us to infer that the complexity of sentence structure influences the gating results. In Table 4, we present one sentence containing multiple clauses that are frequently processed by the top-2 experts.

**Question and Answer.** The percentage of top-2 tokens in question and answer tasks fluctuates across layers (Figure 2c). First, adaptive gating pays extra attention to the question itself. Words listed in Table 4 are some common examples. These tokens often either specify the scope of the question or pose constraints to the answers. Second, in the context side, tokens routed to two experts are closely related to the question in the input as well. For example, asking a question about numbers and computations would result in top-2 gating on the

numbers and the objects those numbers refer to.

**Summarization.** In summarization, the percentage of tokens using two experts decreases in both encoder and decoder layers (Figure 2d). Based on our analysis of sampled tokens, we identify two patterns for tokens that are likely to be routed to top-2 experts. First, tokens with multiple meanings that rely on both themselves and the surrounding context for their ultimate interpretation. They are often routed to two experts in the shallow layers. Second, pronoun tokens, as understanding their referents is crucial for accurate summarization, use two experts in the deeper layers. This pattern is particularly prevalent in this task. Additionally, certain key tokens (e.g. "in conclusion", "however", "in all") that indicate the beginning the central idea or the main opinion of the context are often sent to two experts together with the following tokens.

**Text completion.** Text completion differs from the previous tasks as it is a decoder-only and autoregressive task. The gating results in text completion are influenced by the current prediction being generated. The focus of tokens changes dynamically based on the current prediction. It is challenging to identify specific types of tokens that consistently receive two experts. When predicting a pronoun, for example, the focus shifts to the names of individuals. Similar patterns can be observed for numbers and dates. Additionally, we find that the percentage of tokens routed to two experts is linked to the length of the current sequence. Longer sequences have a higher percentage of top-2 gating.

**Dialogue response.** Dialogue response, compared to text completion, requires more understanding of the narrative input and the dialogue history. We find that lots of effort are put into processing dialogue history. First, one key distinction is that tokens with a conversational meaning occur much more frequently. These words lack informative content but serve to express human-like sentiments, such as gratitude and politeness. We infer that routing these tokens for two experts indicates that there is a difference between the conversational usage and written text and it is also critical to learn where and when these words should be used. Second, given the nature of the dialogue, many conversations are based on underlying assumptions and conditions. Related tokens are usually processed with two tokens to improve the understanding of the context. For instance, the dialogue example provided in Table 4 is built on top of a scenario assuming that

| Task | Top-2 gating tokens |
|---|---|
| Sentiment analysis | realistic, thoroughly, handsome but unfulfilling, simply, is not the worst movie of the year, generic |
| Translation | I believe that anyone who has had the opportunity to visit Algeria during recent months or years can make a better assessment of what this terrible outbreak of terrorism means to the Algerian people and, indeed, I believe that it would be to our credit if we dealt with this issue in an urgent debate. |
| Question and Answer | Which entity, who else, after what, Up until, who was blamed, in terms of, after, Who's death caused this protest? |
| Summarization | Japanese actress Rinko Kikuchi walks Anjali Rao through the streets of Tokyo. She stunned global cinema audiences with her controversial and Oscar-nominated performance as a lonely deaf girl in the film "Babel". Rinko Kikuchi is one of Japan's hottest young actresses and models, recently working with Karl Lagerfeld as the new face of Channel. Despite her success, she remains an unconventional figure in Japan, at odds with the traditional demure image of the Japanese woman and forging a career on her own terms... |
| Text completion | Harris announced he would be stepping down as rabbi in 2011, and the synagogue hired Boris Dolin as his successor. Born and raised in Oregon, Dolin had worked at Temple Beth Israel as a teacher and youth group adviser from 1999 to 2001. |
| Dialogue response | exactly, definitely, hmm, um, well, I guess, [Narrative] Johnathan plans to tell his parents that he is gay. He feels anxious because he doesn't know they will react. He is worried that they will be disappointed or even angry with him. |

Table 4: Examples of tokens using top-2 experts in different tasks. Underlined tokens use top-2 gating in a sequence.

"Johnathan tells his parents that he is gay" and asks the model to answer questions with this condition.

## 4.6 Ablation Study

**Threshold $T$ in adaptive gating.** We now conduct an ablation study on the threshold $T$ introduced in adaptive gating. Increasing the threshold value results in a less sparse model, where more tokens are assigned to the top-2 gating mechanism, subsequently increasing the computational FLOPs. Table 5 shows the inference performance of different tasks when the threshold is increased from 0.05 to 0.5. When using a small threshold of 0.05, both the training time and inference performance closely resemble those of top-1 gating MoE. On the other hand, setting the threshold to 0.4 does not always lead to the same performance as top-2 gating. Together with Table 3, we discover that threshold values of 0.1 and 0.2 often strike a favorable balance between training time and inference performance.

**Curriculum learning.** Essentially, we disable the data order adjustment before each epoch and use the random data loader to feed the training set. We present the performance degradation compared to the full-set adaptive gating in Table 6. Since it is highly possible that there is at least one token that are routed to top-2 experts, the step time of each

| Task | Norm. Training Time | | | | Inference Performance | | | |
|---|---|---|---|---|---|---|---|---|
| | 0.05 | 0.2 | 0.3 | 0.4 | 0.05 | 0.2 | 0.3 | 0.4 |
| Sentiment analysis | 1.02x | 0.77x | 0.92x | 1.01x | 0.912 | **0.918** | 0.917 | 0.918 |
| Translation | 0.88x | 0.83x | 0.83x | 0.88x | 40.2 | **41.1** | 40.8 | 41.1 |
| Question and Answer | 0.92x | 0.87x | 0.93x | 0.96x | 74.3 | **77.6** | 77.6 | 77.6 |
| Summarization | 0.98x | 1.02x | 1.05x | 1.04x | 40.8 | 42.3 | **43.1** | 43.1 |
| Text generation | 0.95x | 0.93x | 0.99x | 1.01x | 16.6 | 17.2 | **17.4** | 17.4 |
| Dialogue response | 0.93x | 0.91x | 1.01x | 1.01x | 12.2 | 12.8 | 13.2 | **13.4** |

Table 5: Overall performance when the threshold $T$ changes. Training time is normalized with reference to top-2 gating MoE. We highlight the best one with the least training time.

| Task | Training Time Inflation | Inference Performance |
|---|---|---|
| Sentiment analysis | 22% | +0.00 |
| Translation | 14% | -0.14 |
| Question and Answer | 9% | -0.21 |
| Summarization | 14% | -0.14 |
| Text completion | 12% | -0.01 |
| Dialogue response | 11% | -0.19 |

Table 6: Overall performance comparison of adaptive gating when data batch is not adjusted.

iteration cannot achieve the same level of reduction as the computation FLOPs. Consequently, the end-to-end training time is significantly inflated, with an average increase of 13.7%. Additionally, the idea of the curriculum also contributes to the improvement in inference performance. The maximum drop is 0.21 in Question and Answer task when the data is fed and trained in a random manner.

## 5 Limitation

**Choice of $k$.** Adaptive gating in MoE currently is limited to top-$k$ gating, where $k$ can be either 1 or 2. This is built on the common practice in extensive prior work that top-2 gating shows a promissing resut in MoE. Further evaluation is necessary to validate the performance of a wider range of $k$ values. Our experiments were conducted on a diverse set of NLP tasks and datasets, but it is essential to note that the effectiveness and efficiency of adaptive MoE may vary depending on the specific task characteristics. Different tasks may exhibit distinct patterns and complexities, which can impact the performance and generalizability of the proposed approach. Further investigation and evaluation on a wider range of tasks would provide a more comprehensive understanding of the limitations and applicability of adaptive MoE.

## 6 Conclusion

This paper demonstrates the effectiveness and flexibility of adaptive gating in MoE models for a wide range of natural language processing tasks. By dynamically adjusting the number of experts based on token characteristics, we achieve improved training efficiency without compromising inference performance. Additionally, the integration of curriculum learning allows us to tackle the challenge of varying execution times, thereby reducing training costs. Our research sheds light on the trade-off between training efficiency and model performance in sparse and dynamic MoE networks, offering valuable insights for the development of more scalable and adaptable language models.

## Acknowledgement

We thank the anonymous EMNLP'23 reviewers and Area Chairs for their constructive and valuable comments. This work was supported in part by funding from the Research Grants Council of Hong Kong (N_CityU139/21, C2004-21GF, R1012-21, R6021-20F, GRF 11209520, and CRF C7004-22G).

## Ethics Statement

There is no ethic problem in this work.

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
