# OpenReview forum: "Adaptive Gating in Mixture-of-Experts based Language Models"
_EMNLP/2023/Conference — EMNLP 2023 Main_

### Official Review · Reviewer_P1xP · 2023-08-03

**Soundness:** 3

**Excitement:**

3: Ambivalent: It has merits (e.g., it reports state-of-the-art results, the idea is nice), but there are key weaknesses (e.g., it describes incremental work), and it can significantly benefit from another round of revision. However, I won't object to accepting it if my co-reviewers champion it.

**Paper Topic And Main Contributions:**

Authors study an important direction of use of mixture of experts (MoE) / switch layers at the FF layers and try to optimize the use of top-2 experts (MoE) vs only top-1 expert (switch). As expected, using top-2 experts leads to better performance than top-1 at the expense of twice as many FLOPs and longer runtime. In this work authors, propose to use top-2 experts only when the difference between the highest expert probability and the second highest expert probability is smaller than a pre-decided threshold saving them compute and retaining the performance of top-2 experts. Unfortunately, to gain full computational speedups they also require a sophisticated curriculum learning schedule.

**Questions For The Authors:**

1) line 365: it was not clear how you initialize the experts. As you say use runs use same parameters, I am assuming the effective FF dimension of each expert is original FF dimension/ 16. If yes, this raises the question how do you split the parameters of original FF layer among these 16 experts. Do you initialize them based by splitting the original init across this in order or do you init the params of new expert FFs randomly. I'd be surprised that you match the performance of original dense model even after throwing away most of the original weights.

2) Table 3: it is not convincing that you train top-1 for longer as the numbers are so close that maybe, if you would have kept a long enough wallclock time for all runs, top-1 would have been comparable to the rest irrespective of the loss.

**Reasons To Accept:**

1. MoE/ switch layers is a useful technique and optimizing it is important - the proposed solution is simple and intuitive.
2. I liked the analysis of complexity of various tokens.
3. The speedups over top-2 are decent, albeit with the use of curriculum.

**Reasons To Reject:**

1. Require sophisticated curriculum making the training pipeline complex and not plug-n-play.
2. The decision of training top-1 baseline for longer is not convincing given that all results are in same ballpark.

**Reproducibility:**

4: Could mostly reproduce the results, but there may be some variation because of sample variance or minor variations in their interpretation of the protocol or method.

**Reviewer Confidence:**

4: Quite sure. I tried to check the important points carefully. It's unlikely, though conceivable, that I missed something that should affect my ratings.

**Typos Grammar Style And Presentation Improvements:**

line 124-125
table1 : top-1 missing x
line 216: what do you mean 55% - is this w.r.t. to a threshhold

---

> ### Author Rebuttal · Authors · 2023-08-28
>
> We thank you for your comments.
> ## Response to the reasons to reject
> 1. We agree that a more sophisticated curriculum design could further improve training efficiency. Currently, we consider the tokens that are handled by two experts as more challenging ones. Therefore, we use a vector that includes the ratio of tokens in each layer receiving top-2 gating to represent the complexity of the data sample.
> 2. We agree that top-1 gating in training is not a common practice. We could remove the training top-1 gating baseline results.
> ## Response to the questions for authors
> 1. We init the parameters of new expert randomly.
> 2. There are at least 55% of all the tokens in the dataset that receive biased top-1 gating decision.

---

### Official Review · Reviewer_ggKr · 2023-08-05

**Soundness:** 3

**Excitement:**

3: Ambivalent: It has merits (e.g., it reports state-of-the-art results, the idea is nice), but there are key weaknesses (e.g., it describes incremental work), and it can significantly benefit from another round of revision. However, I won't object to accepting it if my co-reviewers champion it.

**Paper Topic And Main Contributions:**

The paper proposed an adaptive gating mechanism in a Mixture of Experts (MoE) training scheme, which enables tokens to be processed by a flexible number of experts. The proposed training strategy allows tokens to be processed by a variable number of experts based on expert probability distribution. Overall the results show that the method often achieves improved training efficiency without compromising inference performance, with the integration of curriculum learning to be beneficial for reducing training costs. The authors present an interesting analysis where they explain how the gating mechanism adjusts to each NLP task. For instance, for sentiment analysis the tokens expressing neutral opinion are routed to two experts, while this happens in QA for both questions and context text snippets.

**Questions For The Authors:**

- How are findings in Section 3.1 / Figure 1 drawn? How are these experiments performed in detail?


**Reasons To Accept:**

- The paper is clear and easy to follow. The experimental setup seems solid, with multiple NLP tasks evaluated.
- The proposed approach looks promising, as it usually outperforms baselines in terms of training time, inference FLOPs and performance. The analysis and ablation study nicely explains model design choices and provides insights on how the method works in each NLP task.

**Reasons To Reject:**

- I am skeptical about the novelty of this paper. Despite its potential impact and value, it looks like a combination of existing techniques.
- The performance improvement (training, inference and test performance) over the baseline is quite small.

**Reproducibility:**

4: Could mostly reproduce the results, but there may be some variation because of sample variance or minor variations in their interpretation of the protocol or method.

**Reviewer Confidence:**

2: Willing to defend my evaluation, but it is fairly likely that I missed some details, didn't understand some central points, or can't be sure about the novelty of the work.

**Typos Grammar Style And Presentation Improvements:**

- The authors could use 1-2 sentences in the intro to briefly explain what top-1 and top-2 gating is (L67).
- Section 2 should be changed structure-wise. It has only a single subsection (2.1), while Related Work is just a \paragraph{}.
- Section 4 is also lacking structure. 4.1 is only one sentence.
- The words 'significantly' should be used more carefully (e.g. L 583), only if a significance test is performed.

---

> ### Author Rebuttal · Authors · 2023-08-28
>
> Thank you very much for your comments.
> ## Response to reasons to reject
> 1. To the best of our knowledge, no existing work comprehensively evaluates the performance of adaptive gating in the scope of MoE and the trade-off between the computations per token and the model performance. We provided a variety of experiments with different tasks and models and an in-depth analysis of our findings.
> 2. The performance in Table 3 shows that the FLOPs could be reduced by an average of 1.3x compared with the most widely adopted Top-2 gating while maintaining a comparable inference performance.
> ## Response to questions for the authors
> We will add the experiment details to our paper. The experiments in Figure 1 are measured with the first task in Table 2, that is, using BERT-Base model to perform sentiment analysis tasks using SST-2 dataset. We present four sampled tokens and collect the gating decision of these tokens in a sample iteration during the training stage.
> The model in Table 1 is the same task as Figure 1 and the statistics are profiled with PyTorch Profiler to obtain the time of all the computation kernels in the MoE layer.

---

### Official Review · Reviewer_SPuz · 2023-08-11

**Soundness:** 3

**Excitement:**

2: Mediocre: This paper makes marginal contributions (vs non-contemporaneous work), so I would rather not see it in the conference.

**Paper Topic And Main Contributions:**

This paper introduces adaptive gating in MoE, a flexible training strategy that allows tokens to be processed by a variable number of experts based on expert probability distribution.  Extensive experiments on diverse NLP tasks show that adaptive gating reduces at most 22.5% training time while maintaining inference quality.



**Reasons To Accept:**

The paper is well-written and easy to follow. The illustration of motivation and method is clear.
The design of adaptive gating in MoE is intuitive and technically novel.

**Reasons To Reject:**

This paper uses an existing method and does not mention or lack description of its own.
Insufficient experiments, as many figures as possible should be used to illustrate the author's point of view.
More baseline models are encouraged to be used to evaluate Training Time, FLOPs and Inference Performance.

**Reproducibility:**

4: Could mostly reproduce the results, but there may be some variation because of sample variance or minor variations in their interpretation of the protocol or method.

**Reviewer Confidence:**

2: Willing to defend my evaluation, but it is fairly likely that I missed some details, didn't understand some central points, or can't be sure about the novelty of the work.

**Typos Grammar Style And Presentation Improvements:**

"Figures 1 depict the normalized activation values of four sampled tokens across 16 experts." See line 209, page3.
Figure 2 is not clear.

---

> ### Author Rebuttal · Authors · 2023-08-28
>
> Thank you for your comments.
> To the best of our knowledge, there is no existing work on adaptively controlling the number of experts that each token is being processed. We use various experiments to evaluate the trade-off between the computation per token and model performance and argue that quite a significant portion of computations are actually unnecessary. Besides, we have tried to cover a variety of task types and model types to validate this idea. Could you elaborate on the additional experiments necessary to support our solution?

---

### Meta-Review · Area_Chair_1dGw · 2023-09-07

**Recommendation:** 4

**Metareview:**

The reviewers overall agree that the paper is sound and moderately exciting. While the paper has some flaws, these are mostly because it is difficult to experiment with MoEs if one does not have large computational resources. The initial results are very promising, and the paper is well-written and technically novel.

---

### Decision · Program_Chairs · 2023-10-07

**Decision:**

Accept-Main

**Comment:**

The reviewers overall agree that the paper is sound and moderately exciting. While the paper has some flaws, these are mostly because it is difficult to experiment with MoEs if one does not have large computational resources. The initial results are very promising, and the paper is well-written and technically novel.